# SARS-CoV-2 Positive Hospitalized Cancer Patients during the Italian Outbreak: The Cohort Study in Reggio Emilia

**DOI:** 10.3390/biology9080181

**Published:** 2020-07-22

**Authors:** Carmine Pinto, Annalisa Berselli, Lucia Mangone, Angela Damato, Francesco Iachetta, Marco Foracchia, Francesca Zanelli, Erika Gervasi, Alessandra Romagnani, Giuseppe Prati, Stefania Lui, Francesco Venturelli, Massimo Vicentini, Giulia Besutti, Rossana De Palma, Paolo Giorgi Rossi

**Affiliations:** 1Medical Oncology Unit, AUSL-IRCCS of Reggio Emilia, Viale Risorgimento 80, 42123 Reggio Emilia, Italy; berselli.annalisa@ausl.re.it (A.B.); damato.angela@ausl.re.it (A.D.); iachetta.francesco@ausl.re.it (F.I.); zanelli.francesca@ausl.re.it (F.Z.); gervasi.erika@ausl.re.it (E.G.); romagnani.alessandra@ausl.re.it (A.R.); prati.giuseppe@ausl.re.it (G.P.); lui.stefania@ausl.re.it (S.L.); 2Epidemiology Unit, AUSL-IRCCS of Reggio Emilia, Via Amendola 2, 42122 Reggio Emilia, Italy; mangone.lucia@ausl.re.it (L.M.); venturelli.francesco@ausl.re.it (F.V.); vicentini.massimo@ausl.re.it (M.V.); giorgirossi.paolo@ausl.re.it (P.G.R.); 3Department of Medical Biotechnologies, University of Siena, Strada delle Scotte 4, 53100 Siena, Italy; 4Informatic Technology and Telematics Unit, AUSL-IRCCS of Reggio Emilia, Viale Risorgimento 80, 42123 Reggio Emilia, Italy; foracchia.marco@ausl.re.it; 5Radiology Unit, AUSL-IRCCS of Reggio Emilia, Viale Risorgimento 80, 42123 Reggio Emilia, Italy; besutti.giulia@ausl.re.it; 6Clinical and Experimental Medicine PhD Program, University of Modena and Reggio Emilia, Largo Del Pozzo 71, 41125 Modena, Italy; 7Department of Hospital Care, Emilia Romagna Region, Viale Aldo Moro 21, 4017 Bologna, Italy; rossana.depalma@regione.emilia-romagna.it

**Keywords:** COVID-19, SARS-CoV-2, cancer patients, lung cancer, immunotherapy

## Abstract

In the coronavirus disease (COVID-19) pandemic, cancer patients could be a high-risk group due to their immunosuppressed status; therefore, data on cancer patients must be available in order to consider the most adequate strategy of care. We carried out a cohort study on the risk of hospitalization for COVID-19, oncological history, and outcomes on COVID-19 infected cancer patients admitted to the Hospital of Reggio Emilia. Between 1 February and 3 April 2020, a total of 1226 COVID-19 infected patients were hospitalized. The number of cancer patients hospitalized with COVID-19 infection was 138 (11.3%). The median age was slightly higher in patients with cancers than in those without (76.5 vs. 73.0). The risk of intensive care unit (ICU) admission (10.1% vs. 6.7%; RR 1.23, 95% Confidence Interval (CI) 0.63–2.41) and risk of death (34.1% vs. 26.0%; RR 1.07, 95% CI 0.61–1.71) were similar in cancer and non-cancer patients. In the cancer patients group, 89/138 (64.5%) patients had a time interval >5 years between the diagnosis of the tumor and hospitalization. Male gender, age > 74 years, metastatic disease, bladder cancer, and cardiovascular disease were associated with mortality risk in cancer patients. In the Reggio Emilia Study, the incidence of hospitalization for COVID-19 in people with previous diagnosis of cancer is similar to that in the general population (standardized incidence ratio 98; 95% CI 73–131), and it does not appear to have a more severe course or a higher mortality rate than patients without cancer. The phase II of the COVID-19 epidemic in cancer patients needs a strategy to reduce the likelihood of infection and identify the vulnerable population, both in patients with active antineoplastic treatment and in survivors with frequently different coexisting medical conditions.

## 1. Introduction

An outbreak of novel coronavirus disease (COVID-19) occurred in Wuhan, a city in central China, in December 2019. This disease, characterized by rapid human transmission, has been related to the severe adult respiratory syndrome coronavirus 2 (SARS-CoV-2). Since the first Italian SARS-CoV-2 infection case was reported on 21 February 2020 in Codogno (Lodi), the outspread of the disease particularly affected Italy, which became the main outbreak site in Europe [1,2,3,4,5,6]. The first case of COVID-19 in the United States was reported in January 2020 in Snohomish, Washington, in a patient who had traveled to Wuhan [7]. Ever since, the infection has been spreading and increasing worldwide. The World Health Organization (WHO) first declared COVID-19 a “Public Health Emergency of International Concern”, and on 11 March a “pandemic” [8].

In Italy, the data reported by the Ministry of Health on 12 April showed 156,363 infection cases, 27,847 symptomatic hospitalized patients, of which there were 3343 admitted to intensive care units (ICU), 19,899 deaths, and 34,211 recoveries. The Emilia Romagna Region is the second Italian area by number of infections with 20,098 total cases; within this region, the Province of Reggio Emilia had 3849 cases reported [9].

In the COVID-19 pandemic, cancer patients could be a high-risk group due to their immunosuppressed status and vulnerability to infection resulting from their disease and/or the oncological treatments [10]. The susceptibility of cancer patients to the influenza virus has already been reported. Oncological patients infected with the influenza virus have shown four times higher risk of hospitalization for respiratory distress, and the risk of death is ten times higher than in patients without cancer [11]. Therefore, cancer patients may have higher risks of SARS-CoV-2 infection, and during the infection, a more severe prognosis. In this context, we have no definitive evidence yet. As of today, we have available data for the United States population, after the early reports published in the first weeks of the epidemic on the Chinese population, which differs from the western population in epidemiological, genetic, and clinical characteristics [12,13,14,15,16,17,18,19].

Considering all of the above, there is a high demand for more homogenous data in order to choose the timing and type of antineoplastic treatments, reducing the risk of care disruptions related to social distancing requirements, and manage the appropriate allocation of health care. This information will also be necessary to select which clinical trials to prioritize among the others, and which ones to suspend [17,18,19,20,21,22,23].

This paper reports the results of a cohort study comparing the outcomes of patients admitted to the Provincial Hospital of Reggio Emilia for COVID-19 according to the oncological history. We also compare the standardized COVID-19 hospitalization ratio of cancer patients with the general population and describe characteristics of cancer patients in the cohort.

## 2. Methods

### 2.1. Study Design

This is an Observational cohort Study performed in the Hospitals in the province of Reggio Emilia (530,000 inhabitants), which has designated COVID-19 Units in four sites (Reggio Emilia City, Castelnovo Monti, Guastalla and Scandiano). All hospitalizations of patients with COVID-19 disease were identified between 1 February and 3 April 2020 through the hospital electronic system. Cases were followed up for ICU admission and death until 30 June 2020. Epidemiological, clinical, laboratory, and radiological data were collected and evaluated from the electronic medical records and the Reggio Emilia Tumor Registry archive. This study was approved by the local Ethics Committee of the Area Vasta Emilia Nord (AVEN) (Study No. 281/2020/OSS/IRCCSRE approved on 24 March 2020).

### 2.2. Definitions

A COVID-19 patient was diagnosed based on real-time reverse transcription polymerase chain reaction (RT-PCR) of nasopharyngeal specimens according to WHO indications [24]. The presence of SARS-CoV2 related pneumonia was assessed through computer tomography (CT) according to the typical findings of mostly peripherally distributed ground glass opacities (GGO) and consolidations.

A cancer patient, excluding non-melanoma skin cancers, was defined if the patient was in the Reggio Emilia Tumor Registry archive; for diagnoses that occurred in the last two years, for which the cancer registry is not complete, the Oncology Unit electronic clinical records were checked.

### 2.3. Imaging Evaluation

Lung CT scans were performed using one of three scanners (128-slice Somatom Definition Edge, Siemens Healthcare; 64-slice Ingenuity, Philips Healthcare; 16-slice GE Brightspeed, GE Medical System) without contrast media injection, with the patient in supine position, during end-inspiration. Scanning parameters were: tube voltage 120 KV, automatic tube current modulation, collimation width 0.625 or 1.25 mm, acquisition slice thickness 2.5 mm, and interval 1.25 mm. Images were reconstructed with a high-resolution algorithm at slice thickness 1.0/1.25 mm. During CT reporting, each radiologist completed both the usual radiology report as well as a structured report, including the presence/absence of GGO and consolidations, and the extension of pulmonary lesions using a visual scoring system (0%, 1–19%, 20–39%, 40–59%, and ≥60% of parenchymal involvement) [25]. 

### 2.4. Statistical Analysis

For descriptive analysis continuous variables were presented in median and range, while categorical variables were presented as frequencies and proportions. Pearson chi-squared test were implemented to test sex ratio differences between groups. Age and sex standardized hospitalization ratio (SHR) was also calculated with indirect standardization method to compare the risk of hospitalization for COVID-19 in people 0 to 84 years old with a previous diagnosis of cancer and the general population, and denominators were estimated according to data on cancer prevalence by age from 0 to 84, as reported by the Reggio Emilia cancer registry [26]. To compare intensive care unit (ICU) admission and mortality between SARS-CoV-2 infected cancer and non-cancer patients, sex and age adjusted odds ratios (OR) and the corresponding 95% confidence interval (95% CI) were calculated using multivariate logistic regression models. No formal statistical test of hypothesis has been conducted, 95% CI are reported to show precision of estimates and the certainty of the association. Odds ratio of death and ICU admission at 40 days since hospital admission in oncologic patients were also calculated by means of multivariate logistic regression analysis, adjusting for age, gender, metastatic disease, and time since cancer diagnosis. Computed tomography (CT) findings were compared between cancer and non-cancer patients by means of chi-squared test, and odds ratios (OR) for each CT finding (ground–glass opacities (GGO), consolidations, and visual extension of pulmonary lesions ≥40% and ≥60%) were calculated by means of multivariate logistic regression analysis, adjusting for age and sex. All statistical analysis was performed using SPSS Statistics version 26.0. 

## 3. Results

Between 1 February and 3 April 2020, a total of 1226 COVID-19 infected patients were admitted at the Provincial Hospital of Reggio Emilia (Table 1), representing the 0.23% of the resident population. The gender distribution was 493 (40.2%) females and 733 (59.8%) males. The median age of all patients was 73 years (range 23–100), ICU admission was needed for 92 (7.5%) patients, and 330 patients (26.9%) died. 

The number of cancer patients hospitalized with SARS-CoV-2 infection was 138 (11.3%) for a standardized hospitalization ratio compared with the general population—98 (95% CI 73–131). There were no small differences or no differences between the hospitalized patients with and without cancer by median age (76 vs. 73 years), risk of ICU admission (10.1% vs. 6.7%; OR 1.23, 95% CI 0.63–2.41), and fatality rate (34.1% vs. 26.0%; OR 1.07, 95% CI 0.71–1.61). The percentage of patients with a cancer diagnosis occurred more than 5 years before the COVID-19 hospitalization was 64.5% (89/138), 23.9% (33/138) had a diagnosis from 5 to ≥1 years, and 11.6% (16/138) had the cancer diagnosis in the last year. The five most frequent tumor sites were: prostate cancer (30 patients), breast cancer (27 patients), colorectal cancer (25 patients), bladder cancer (12 patients), and lung cancer (9 patients). Patients with a period of time >5 years between cancer diagnosis and COVID-19 hospitalization had a higher median age (79 years; range 57–98) and more frequently had one or more comorbidities (chronic obstructive pulmonary disease in 13.5%, hypertension in 79.8%, cardiovascular disease in 62.9%, and diabetes mellitus in 31.5%) compared with patients with a more recent cancer diagnosis (Table 2, Figure 1). 

Baseline lung CT scan was available in 1015/1226 patients (105/140 cancer patients and 910/1086 non-cancer patients). Lung CT scans demonstrated ground–glass opacities in 98 (95%) cancer patients and 851 (94%) non-cancer patients, and consolidation in 76 (74.5%) cancer patients and 619 (68.3%) non-cancer patients. Cancer patients had slightly higher disease extension by means of visual score, with 25 (28.1%) vs. 162 (20.8%) with ≥60% extension and 12 (13.5%) vs. 175 (22.4%) with <20% extension, but differences are compatible with random fluctuations (Table 3).

The OR of death and ICU admission, adjusted for age, gender, metastatic disease, and time since cancer diagnosis, at 40 days since hospital admission in the 138 cancer patients were reported in Table 4. In this model, male gender (OR 1.98, 95% CI 0.87–4.51), age > 74 years (OR 2.95, 95% CI 1.29–6.78), metastatic disease (OR 3.84, 95% CI 0.80–17.51), bladder cancer (OR 1.80, 95% CI 0.48–4.75), and cardiovascular disease (OR 1.58, 95% CI 0.68–3.68) were associated with mortality risk in cancer patients. Having a recent cancer diagnosis, COPD, and diabetes were associated with lower mortality, while former smokers had higher mortality than smokers and never smokers in this subpopulation. Male gender (OR 1.66, 95% CI 0.77–3.55), metastatic disease (OR 2.14, 0.50–9.16), and bladder cancer (OR 1.52, 95% CI 0.42–5.47) were associated with an overall OR of death or ICU admission.

A total of 14 (1.1%) cancer patients had ongoing oncological treatment within 60 days before the COVID-19 infection hospitalization (Table 5). A lung involvement was detected in 9 (64.3%) of these patients (3 primary tumors and 6 lung metastases). Bladder cancer (3 patients) and ovarian cancer (3 patients) were the other most two frequent tumor types. Most of the patients had metastatic disease (78.6%). Chemotherapy was administered in 12 (85.7%) cases (alone in 8 patients, in combination with antiangiogenic drugs in 4 patients and with immunotherapy in 1 patient). Two patients underwent hormone therapy (tamoxifen in 1 breast cancer patient and leuprolide plus enzalutamide in 1 prostate cancer patient). The treatment of COVID-19 infection for those patients was: hydroxychloroquine and/or azithromycin in the 92.8%, antiviral drugs in the 50%, and glucocorticoids and heparin in the 85.7%. Fatal events occurred in 5/14 (35.6%) in all-male patients (2 colorectal cancer, 2 bladder cancer, and 1 non-small cell lung cancer). The median age of patients was 79 years (range 65–81). Of the 5 deceased patients, 4 were current/former smokers, 3 were taking medical therapy for hypertension, and only one had been vaccinated for influenza virus. Lastly, 4/5 (80%) patients had a metastatic disease and underwent to chemotherapy; specifically, 2 patients with chemotherapy alone and 2 patients with chemotherapy plus antiangiogenic therapy. 

## 4. Discussion

Considering the estimated prevalence of resident people who had a diagnosis of cancer in their life, we found a similar hospitalization rate for COVID-19 compared with the general population (SHR 98, 95% CI 73–131), suggesting small, if any, differences in susceptibility to infection. Surprisingly, within hospitalized COVID-19 patients, we found no excess of ICU admission and deaths in patients with cancer compared with patients without cancer. This finding is consistent with the results obtained in a cohort of patients with COVID-19 in the same area, that showed a small excess of death (HR 1.4), but also a small excess of being hospitalized (HR 1.4), suggesting that, once hospitalized, there should not be any excess of death [27]. Among cancer patients only having a metastatic disease, there was a strong determinant of mortality, together with age and male gender; a small increase was also observed for cardiovascular disease. On the contrary, having a recent diagnosis, diabetes, and COPD were associated with lower mortality in this cohort of hospitalized COVID-19 patients with cancer, suggesting that probably these conditions favor hospitalization of patients even when COVID-disease is less severe or life-threatening. 

Among the limits of this study, the small number of cancer patients affects the precision of our estimates, particularly for the comparisons within cancer patients. Another limit is that we only included hospitalized patients, thus we could not study the effect of having had a cancer on the probability of infection and mortality. Furthermore, limiting the cohort to hospitalized patients may introduce a collider bias, in fact some conditions, in particular comorbidities, may increase the probability of being hospitalized in COVID-19 patients. Therefore, their effect on mortality can be masked or even reverted, when comparing the mortality of people with the comorbidity and those without the comorbidity, conditional that both have been hospitalized. 

Some early reports on the Chinese population suggest that cancer patients are at higher risk of SARS-CoV-2 infection and could have higher morbidity and mortality than the general population. In a nationwide analysis that collected 1590 COVID-19 cases from 575 Chinese hospitals, 18 (1%) patients had a history of cancer, which seems to be higher than the cancer incidence in China of 285.83 per 100,000 people (0.29%). Cancer diseases were associated to a higher risk of severe events, defined as admission to the ICU or death, seen in 7/18 (39%) patients with cancer compared with 124/1572 (8%) patients without cancer (*p* = 0.0003) [12]. In another series from a single institution (Zhongnan Hospital of Wuhan University), the infection rate of SARS-Cov-2 in patients with cancer was 0.79%, which was higher than the cumulative incidence of all diagnosed COVID-19 cases in the city of Wuhan over the same time period (41,152 cases on 11,081,000 residents, 0.37%) [13]. In the retrospective cohort study, which included 28 COVID-19 infected cancer patients from three hospitals in Wuhan, 15 (53.6%) patients had severe events and mortality was 28.6% [14]. The Chinese Center for Disease Control and Prevention reported the characteristics of 72,314 COVID-19 cases. In the 107 (0.5%) cancer patients, the mortality was 5.6%, higher than the overall reported deaths from COVID-19 (2.3%) [15]. The WHO-China Joint Mission on COVID-19 also reported higher mortality among patients with pre-existing cancers (7.6%) than patients without comorbidity (1.4%) [16]. These comparisons were not age-adjusted. A multi-center study performed in 14 hospitals of Hubei Province evaluated 105 patients with cancer and 536 without cancer. Compared with COVID-19 patients without cancer, cancer patients had higher rates of ICU admissions (OR 2.84; *p* < 0.01) and deaths (OR 2.34; *p* = 0.03) [17]. 

A total of 167 cases of COVID-19 were found in the United States in a long-term care facility in King Country, Washington (101 residents, 50 health care personnel, and 16 visitors). Most facility residents (94.1% of 101) had a chronic underlying health condition, which was cancer in 15 (14.9%) [28]. Among the 393 patients with confirmed COVID-19 admitted between 5 March (date of the first positive case) and 27 March 2020 at a referral center and community hospital of Manhattan, New York City, 23 (5.9%) were cancer patients. The cancer patients underwent mechanical and non-mechanical ventilation in 7.7% and 4.9% of cases, respectively [29]. A total of 164 patients with COVID-19 infection and solid tumors were treated in Montefiore Health Hospital System in New York, from 18 March to 8 April 2020. The mortality was 25% for all patients suffering of solid tumors. Active chemotherapy and radiation therapy treatment were not associated with increase case fatality [18]. In a subsample of 355 patients with COVID-19 who died in Italy, 87 (24.5%) patients had active cancer [30].

In the Reggio Emilia Study, 138 (11.3%) of 1226 hospitalized COVID-19 patients had a history of cancer. Comparing all hospitalized COVID-19 cancer patients with patients without cancer, there were small or no differences in median age (76 vs. 73 years), admission to the ICU (10.1% vs. 6.7%), fatal events (34.1% vs. 26.0%), and in the severity of lung scan CT findings. These data are different from those described in the reports in China, and there could be several reasons for these discrepancies. Generally, the composition by age and the overall/site incidence of tumors are different between the Chinese and Italian populations. The cancer incidence in Italy and in China is 562.54 and 285.83 per 100,000 people, respectively [31,32]. The distribution of COVID-19 infected cases is very different in the two countries: people aged ≥70 years represent 37.6% of cases in Italy and only 11.9% in China [30]. Furthermore, the different potentialities of access to antineoplastic therapies and the availability of regional data from cancer registries must be considered. The Italian data are similar in general characteristics of population, median age, ICU admission, and mortality to those reported in the United States [28,29,33,34,35].

The correlations between cancer immunotherapy and COVID-19 developments have been the subject of several hypotheses, none of which have been conclusive [36,37]. In the multi-center study of Hubei Province 6/105 (5.71%) cancer patients had undergone immunotherapy for lung cancer within 40 days before the onset of COVID-19 symptoms and 2/6 patients (33.3%) had died [17]. Another study that included 69 outpatients with diagnosis of advanced lung cancer and COVID-19 infection, treated at Memorial Sloan Kettering Cancer Center in New York, found no significant association between receipt of prior PD-1 blockade and COVID-19 severity [19]. The Cohort Study of Reggio Emilia in hospitalized COVID-19 infected patients found that only 1/14 (7.1%) cancer patient was undergoing immunotherapy (plus chemotherapy) for advanced malignant pleural mesothelioma. In the clinical records of the Medical Oncology Unit of Reggio Emilia and University of Modena and Reggio Emilia, 337 patients were registered and treated with checkpoint inhibitors between 1 January and 30 April 2020. Among this group of cancer patients, only 3 (0.89%) were hospitalized for COVID-19 (1 metastatic colorectal cancer patient treated with FOLFOXIRI/bevacizumab/atezolizumab, 1 advanced malignant pleural mesothelioma patient treated with cisplatin/pemetrexed/pembrolizumab, 1 metastatic clear cell renal cancer patient treated with ipilimumab/nivolumab), and among them, only 1 (0.30%) died [38]. These early data are promising for the safety of the continued use of checkpoint inhibitors during the COVID-19 epidemic and the next phase II. Furthermore, the observation of lower mortality in cases with recent diagnosis, even if the probably is biased by a higher hospitalization of COVID-19 patients with concomitant diseases as cancer, is supportive for the real impact of cancer and cancer treatments on COVID-19 prognosis.

Another topic of debate concerns the risks and correlations between SARS-CoV-2 pneumonia and lung cancer [39]. Lung cancer patients usually have altered lung function, and therefore they could have of severe forms of COVID-19 infection. Furthermore, the habit of tobacco smoking usually present in lung cancer patients is the main cause of chronic obstructive pulmonary disease, which has been reported as an independent risk factor in severe COVID-19 cases [19,40], but not in Reggio Emilia Province [27]. Reports in the Chinese population regarding a small group of COVID-19 infected patients with cancer have shown that lung cancer was the most frequent type of malignancy: 5 (27.8%) of 18 patients in a nationwide analysis [12], 7 (58.3%) of 12 patients in the Zhongan Hospital series [13], 7 (25.0%) of 28 patients in three hospitals in Wuhan [14], and 22 (20.9%) of 105 in Hubei Hospital series [17]. Among the 157 patients with solid tumors in Montefiore Health System, 11 (7.0%) had a lung cancer and 6/11 (55%) of these died from COVID-19 infection [18]. Univariable analyses in the 200 patients included in the Thoracic Cancers International COVID-19 Collaboration Registry (TERAVOLT Cohort Study), age > 65 years, status of current/former smoker, treatment with chemotherapy, and presence of any comorbidity were associated with increased risk of death. However, in a multivariable analysis, only smoking history (OR 3.18, 95% CI 1.11–9.06) was associated with increased risk of death [41]. In our study regarding the Reggio Emilia Province, 9/138 (6.5%) lung cancer patients were reported and 3 (21.4%) out of 14 cancer patients had undergone an active antineoplastic treatment within 60 days before COVID-19 infection hospitalization. Among the 5 deaths of this group, only one patient was undergoing treatment for lung cancer.

## 5. Conclusions

In the Reggio Emilia Cohort Study, the incidence COVID-19 hospitalization in people who had a previous diagnosis of cancer was similar to that in the general population. In our study, we observed that after hospitalization, cancer patients have a similar risk of ICU and no excess of risk of death compared with patients without cancer. Male gender, age > 70 years, metastatic disease bladder cancer, and cardiovascular disease were associated with mortality risk in these cancer patients. The analysis of clinical data shows no strong associations with previous cancer immunotherapy and antineoplastic treatments for lung cancer. However, our study has limitations related to the retrospective nature, source of information, and size of the sample that could limit the power.

The clinical decisions regarding the beginning, timing, and type of therapy should consider only the biological tumor characteristics and the clinical aspects of the patient. The increasing population of cancer survivors and the increased percentage of cancer survivors over the age of 65 years, with different coexisting medical conditions, needs special attention as public health issue [33], but our data suggest that mainly metastatic cancer patients have really different prognosis for COVID-19. Moreover, the outbreaks of COVID-19 in long-term care facilities, hospices, and home care assistance can have a considerable impact on vulnerable advanced cancer patients and older patients. A public health strategy is needed that considers globally, in all situations, the preventive and care aspects for cancer patients in the COVID-19 emergency and phase II challenges.

## Figures and Tables

**Figure 1 biology-09-00181-f001:**
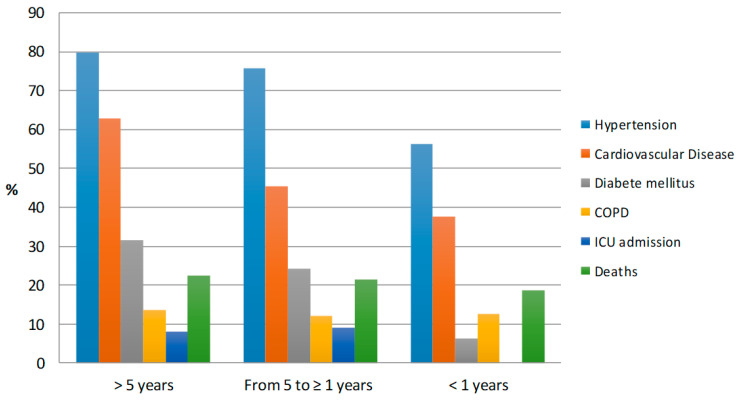
Distribution of comorbidities, intensive care unit (ICU) admission and deaths in cancer patients in subgroups estimated by diagnosis of cancer and COVID-19 (No. 138). COPD = Chronic Obstructive Pulmonary Disease.

**Table 1 biology-09-00181-t001:** Characteristics of 1226 hospitalized coronavirus disease (COVID-19) patients at Reggio Emilia Hospital from 1 January to 3 April 2020.

Characteristics	Total Patients (No. 1226)	Cancer Patients (No. 138)	Non-Cancer Patients (No. 1088)	Adj OR
Sex, No. (%)				
Female	493 (40.2%)	52 (37.7%)	441 (40.5%)	
Male	733 (59.8%)	86 (62.3%)	647 (59.5%)	
Mean age, years (SD)	71.7 (14.5)	76.8 (10.4)	71.0 (14.8)	
Median age, years (range)	73 (23–100)	76 (45–98)	73 (23–100)	
ICU hospitalized, No. (%)	92 (7.5%)	14 (10.1%)	73 (6.7%)	1.23, 95% CI 0.63–2.41
Deaths, No. (%)	330 (26.9%)	47 (34.1%)	283 (26.0%)	1.07, 95% CI 0.71–1.61

SD = standard deviation; OR = Odds Ratio; ICU = Intensive Care Unit.

**Table 2 biology-09-00181-t002:** Time between diagnosis of cancer and COVID-19 hospitalization (No. 138).

	>5 Years	From 5 to ≥1 Years	<1 Year
Patients, No. (%)	89 (64.5%)	33 (23.9%)	16 (11.6%)
Gender, No. (%)			
Female	39 (43.8%)	11 (33.3%)	3 (18.8%)
Male	50 (56.2%)	22 (66.7%)	13 (81.3%)
Mean age years	78.3	75.3	72.5
Median, years (range) age	79 (57- 98)	75 (49–92)	74 (45–89)
Smoking status, No. (%)			
Never	53 (59.6%)	16 (48.5%)	7 (43.8%)
Former/Current	29 (32.5%)	15 (45.4%)	8 (50.0%)
Missing	7 (7.9%)	2 (6.1%)	1 (6.2%)
Tumor site, No. (%)	Breast 20 (22.5%)		
	Colorectal 18 (20.2%)		
	Prostate 16 (18.0%)		
	Bladder 6 (6.7%)		
	Lung 5 (5.6%)	Colorectal 5 (15.2%)	Bladder 4 (25%)
	Kidney 4 (4.5%)	Prostate 10 (30.3%)	Prostate 4 (25%)
	Stomach 4 (4.5%)	Breast 7 (21.2%)	Lung 2 (12.5%)
	Thyroid 4 (4.5%)	Bladder 2 (6.1%)	Colorectal 2 (12.5%)
	Uterus 3 (3.4%)	Lung 2 (6.1%)	Mesothelioma 1 (6.2%)
	Other 9 (10.0%)	Other 7 (21.2%)	Other 3 (15.8%)
Comorbidities, No. (%)			
COPD	12 (13.5%)	4 (12.1%)	2 (12.5%)
Hypertension	71 (79.8%)	25 (75.8%)	9 (56.3%)
Cardiovasc. disease	56 (62.9%)	15 (45.5%)	6 (37.5%)
Diabetes mellitus	28 (31.5%)	8 (24.2%)	1 (6.3%)
ICU admission, No. (%)	11 (12.4%)	3 (9.1%)	0
Deaths, No. (%)	35 (39.3%)	9 (27.3%)	3 (18.8%)

COPD = Chronic Obstructive Pulmonary Disease; ICU = Intensive Care Unit.

**Table 3 biology-09-00181-t003:** Distribution of lung computed tomography (CT) scan findings in cancer and non-cancer patients hospitalized for COVID-19 (No. 1015).

Lung CT Scan Finding		Total Patients (No. 1015)	Cancer Patients (No. 105)	Non-Cancer Patients (No. 910)	*p*	OR (95% CI)
Ground-glass opacities	Presence	98 (95.1%)	98 (95.1%)	851 (94.0%)	0.649	1.44 (0.56–3.71)
	Absence	5 (4.9%)	5 (4.9%)	54 (6.0%)		
	Missing	2	2	5		
Consolidation	Presence	76 (74.5%)	76 (74.5%)	619 (68.3%)	0.200	1.38 (0.86–0.20)
	Absence	26 (25.5%)	26 (25.5%)	287 (31.7%)		
	Missing	3	3	4		
Visual extension	0	0 (0.0%)	0 (0.0%)	12 (1.5%)	0.190	1.48 (0.95–2.33) * 1.39 (0.84–2.29) **
	1–19%	12 (13.5%)	12 (13.5%)	163 (20.9%)		
	20–39%	28 (31.4%)	28 (31.4%)	262 (33.6%)		
	40–59%	24 (27.0%)	24 (27.0%)	181 (23.2%)		
	≥60%	25 (28.1%)	25 (28.1%)	162 (20.8%)		
	Missing	16	16	130		

CT = Computed Tomography; *p* = Pearson’s chi-squared test; OR = Odds Ratio adjusted for age and sex. * For visual extension >39% (40–59% and ≥60% vs. 0%, 1–19%, and 20–39% classes). ** For visual extension ≥60% (≥60% vs. all other classes).

**Table 4 biology-09-00181-t004:** Odds ratio of death at 40 days since hospital admission in cancer patients (No. 138).

	Death (No. 52)		ICU Admission or Death (No. 58)		ICU Admission (No. 12)	
	OR	95% CI	OR	95% CI	OR	95% CI
Age (>74 vs. ≤74 years) *	2.95	1.29–6.78	1.53	0.73–3.23	0.11	0.03–0.41
Age (per 10 year increase) *	2.20	1.41–3.42	1.60	1.09–2.35	0.28	0.13–0.57
Gender (male vs. female) ^	1.98	0.87–4.51	1.66	0.77–3.55	1.85	0.46–7.47
Metastatic disease ^^	3.84	0.80–17.51	2.14	0.50–9.16	-	**
Cancer diagnosis ≤5 years “	0.31	0.11–0.84	0.38	0.15–0.95	0.2	0.03–1.30
Cancer diagnosis <1 years ^§^	0.13	0.02–1.04	0.14	0.02–0.92	1	-
Breast cancer ***	0.74	0.21–2.63	0.90	0.28–2.88	1.41	0.16–12.65
Colorectal cancer ***	0.94	0.34–2.61	1.25	0.48–3.23	1.44	0.31–6.51
Bladder cancer ***	1.80	0.48–4.75	1.52	0.42–5.47	1.52	0.15–15.21
Lung cancer ***	0.78	0.16–3.74	0.66	0.14–3.03	0.95	0.08–11.25
Former smokers ***	1		1		1	
Never smokers *	0.47	0.19–1.25	0.48	0.20–1.18	0.21	0.04–1.07
Smokers *	0.38	0.08–1.32	0.55	0.13–2.30	0.44	0.04–4.52
COPD ***	0.38	0.11–1.28	0.46	0.15–1.42	2.43	0.39–15.18
Hypertension ***	1.74	0.67–4.54	1.21	0.51–2.88	0.96	0.22–4.22
Diabetes mellitus ***	0.56	0.23–1.38	0.75	0.33–1.71	0.62	0.14–2.78
Cardiovasc. disease ***	1.58	0.68–3.68	1.14	0.52–2.49	0.68	0.16–2.86

OR = Odds Ratio; ICU = Intensive Care Unit; COPD = Chronic Obstructive Pulmonary Disease. * Adjusted for sex, presence of metastatic disease, time since cancer diagnosis (5-year cut off). ** None of the 14 metastatic patients have been admitted in ICU. Consequently, all the other variables are only adjusted for age, sex, and time since cancer diagnosis. *** Adjusted for age, sex, presence of metastatic disease, and time since cancer diagnosis. ^ Adjusted for age, presence of metastatic disease, and time since cancer diagnosis (5-year cut off). ^^ Adjusted for age, sex, and time since cancer diagnosis (5-year cut off). “ Adjusted for age, sex, and presence of metastatic disease, ^§^ Adjusted for age, sex, and presence of metastatic disease. (No patients with cancer diagnosis within 1 year from before admission were referred to ICU.

**Table 5 biology-09-00181-t005:** Characteristics of cancer patients with COVID-19 hospitalized and medical oncology active treatment (No. 14).

Characteristics	All Patients	Deaths
Gender, No. (%)		
Female	5 (35.7%)	0
Male	9 (64.3%)	5
Median age, years (range)	79 (49–90)	79 (65–81)
Stage of disease, No. (%)		
Localized	3 (21.4%)	1
Metastatic	11 (78.6%)	4
Setting of therapy, No. (%)		
Adjuvant	3 (21.4%)	1
Metastatic 1st line	4 (28.6%)	1
Metastatic > 1st line	7 (50.0%)	3
Tumor site, No. (%)	Lung 3 (21.4%)	
	Bladder 3 (21.4%)	Lung 1
	Ovary 3 (21.4%)	Bladder 2
	Colorectal 2 (14.3%)	
	Breast 1 (7.1%)	Colorectal 2
	Pleura 1 (7.1%)	
	Prostate 1 (7.1%)	
Lung metastasis, No. (%)	6 (42.8%)	1
Cancer treatments within 60 days, No. (%)		
CHT alone *	8 (57.1%)	3
CHT plus antiangiogenic drugs **	3 (21.4%)	2
CHT plus immunotherapy ***		
Hormone therapy ^§^	1 (7.1%)	0
	2 (14.3%)	0
Smoking status, No. (%)		
Never	8 (5.71%)	3
Former/Current	4 (28.6%)	2
Missing	2 (14.3%)	0
Hypertension, No. (%)	10 (71.4%)	3
Diabetes mellitus, No. (%)	2 (14.3%)	1
COPD, No. (%)	2 (14.3%)	0
Cardiovasc. disease, No. (%)	4 (28.6%)	3
Flu vaccination, No. (%)	6 (42.8%)	1
COVID-19 treatments, No. (%)		
Hydroxychloroquine and/or	13 (92.8%)	5
Azithromycin		
Antiviral drugs ^	7 (50%)	4
Glucocorticoids	12 (85.7%)	4
Heparin	12 (85.7%)	4

CHT = Chemotherapy; COPD = Chronic Obstructive Pulmonary Disease. * Carboplatin plus pemetrexed (no. = 2), paclitaxel (no. = 2), epirubicin (no. = 1), gemcitabine (no. = 1), vinorelbine (no. = 1), capecitabine plus temozolomide (no. = 1). ** Capecitabine plus bevacizumab (no. = 1), FOLFOX plus bevacizumab (no. = 1), docetaxel plus nintedanib (no. = 1). *** Carboplatin plus pemetrexed and pembrolizumab (no. = 1). ^§^ Tamoxifen (no. = 1), enzalutamide plus leuprolide (no. = 1). ^ Remdesevir, lopinavir/ritonavir, alganciclovir.

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
