# Peer review of "SARS-CoV-2 Positive Hospitalized Cancer Patients during the Italian Outbreak: The Cohort Study in Reggio Emilia"

_biology, 2020, doi:10.3390/biology9080181_

Round 1
Reviewer 1 Report
The writing of the manuscript must be improved. There are very much misprints (e.g. line 80: COVID-19 desease) (e.g. line 105: Standardised). (Table 2: Diabete mellitus)
Abstract
Lines 33-35: “In the Reggio Emilia Study, the incidence of hospitalization for COVID-19 seems slightly lower in people with previous diagnosis of cancer than in the general population”.
The authors must provide data on the incidence rate of COVID-19 in people with and without a previous diagnosis of cancer in order to make this claim.
Introduction
Line 43. It is not correct to write “A cluster”.
Cluster in epidemiology is a grouping of cases in a given area, in a particular period without considering whether the number of cases is greater than expected. For this reason, authors should write “An outbreak”
Lines 72-75: “This paper is a retrospective study on the cancer prevalence in COVID-19 hospitalized patients and on the oncological history and outcomes of COVID-19 infected cancer patients admitted to the Provincial Hospital of Reggio Emilia. This is the first cohort study on COVID-19 cancer patients among the European population.”
This paragraph is confusing. The studies that analyse prevalence are the cross-sectional studies. Furthermore, the wording of the objectives of this study is not clear. The authors need to clarify this paragraph.
“This is the first cohort study on COVID-19 cancer patients among the European population” should appear in the Discussion (and not in the Introduction).
Methods
Authors should better specify the design of their study (is a retrospective cohort study???)
Lines 80-82: “All hospitalized patients with COVID-19 desease were identified between February 1st and April 3, 2020 through the hospital electronic system. Cases were followed up for ICU admission and death until April 3, 2020.”
What cases did the authors include? Were the cases prevalent as of February 1 included? Or were only cases with a diagnosis date between February 1 and April 3 included?
It seems that the follow-up ended on April 3rd? Would it not have been more appropriate for the follow-up of the cases studied to end when the patient was discharged or died, regardless of whether this occurred after April 3?
Line 87. “thoracic computerized tomography criteria”.
Authors should specify these criteria.
Before using an abbreviation, authors must specify what it corresponds to:
e.g. Line 88. RT-PCR
Line 89. excluding non-melanoma skin cancers
Why were these patients excluded?
Line 103. “continuous variables were presented in median and range”.
Why was the used the median and range??
Line 105. “Pearson chi-squared test were implemented to test sex ratio differences between groups”.
Pearson chi-squared test does not test for differences in rates. The authors should clarify their sentence.
Line 108. “data on cancer prevalence from Italian cancer registries [ref]”.
The reference is missing. In addition, authors should use data from the cancer registry in the population served by the hospital where the study was conducted.
What method did authors use to obtain the standardized rates? (the direct or indirect one?) Authors should explain this.
Lines 111-114: “Odds Ratio of death and ICU admission at 40 days since hospital admission in oncologic patients were calculated by means of multivariate logistic regression analysis, adjusting for age, gender, metastatic disease, and time since cancer diagnosis (for this analysis, the follow up for death in cancer patients is updated on May 27, 2020).”
Why didn't authors use the Hazard Ratios (HR)??
Why is no follow-up done until the patient dies or is discharged, regardless of the date of these events?
Authors should provide data on the number of people residing in the study area (with the reference that supports such data).
Results
Lines 129-130. “The number of cancer patients hospitalized with SARS-CoV-2 infection was 138 (11.3%) for a standardized hospitalization ratio compared to the general population of 89.9 (95%CI 75.5-106.1).”
What is the standardized hospitalization rate for people with cancer?
What is the standardized hospitalization rate for people without cancer?
Why don't authors describe this rate per 100,000 people?
Lines 131-132. “There were no statistically significant differences between the hospitalized patients with and without cancer by median age (76 vs. 73 years)”
Authors should give the p value.
In addition, in Methods authors should specify which statistical test they used to make such comparison.
Lines 138-139. “Patients with a period of time > 5 years between cancer diagnosis and COVID-19 hospitalization had a higher median age (79 years; range 57-98)”
Authors should give the p value.
In addition, in Methods authors should specify which statistical test they used to make such comparison.
“A COVID-19 patient was diagnosed based on the thoracic computerized tomography criteria and by RT-PCR of nasopharyngeal specimens according to WHO indications (lines 87-88)”.
However, “Baseline lung CT scan was available in 1015/1226 patients (105/140 cancer patients and 910/1086 non-cancer patients). (lines 149-150)”.
If to diagnose COVID-19 it was necessary the thoracic computerized tomography criteria, there is an inconsistency between what is reflected in Methods and what is described in Results.
Line 160-165. “The OR of death and ICU admission, adjusted for age, gender, metastatic disease and time since cancer diagnosis, at 40 days since hospital admission in the 138 cancer patients were reported in Table 4. In this model, male gender (OR 1.96, 95%CI 0.87-4.44), age > 70 years (OR 4.32, 95%CI 1.49-12.55), metastatic disease (OR 4.97, 95%CI 1.24-19.94), bladder cancer (OR 2.1, 95% CI 0.55-7.94) and cardiovascular disease (HR 1.72, 95% CI 0.75-3.96) were associated with mortality risk in cancer patients.”
This paragraph shows confusing data, mixing OR and HR.
Discussion
Lines 202-203. “Considering the estimated prevalence of resident people who had a diagnosis of cancer in their life, we found a slightly lower hospitalization rate for COVID-19 than in the general population”.
With the current information available in the Results of this study, the authors cannot make this claim.
Lines 218-219. “which seems to be higher than the cancer prevalence in China of 285.83 per 100,000 people (0.29%)”.
The authors mix prevalence data with incidence rate data.
In the Discussion the authors provide information on results of various works published by other authors, but they are hardly compared with them.
Are the criteria for ICU admission the same for cancer patients as for non-cancer patients?
24.1% of the patients died. What could be the reasons for such high mortality compared to that found by other authors?
The authors have not described the limitations of this study (e.g. authors have not done the sample size calculation prior to any study. Too small a sample size could have limited the power of the study, and therefore the possibility of finding significant differences). (e.g. Retrospective nature of the study, source of information used (electronic medical records)…)
References
Not all references meet the standards of the Journal. This section should be reviewed carefully.
e.g. Reference 39. Its volume (382) and page numbers (1708-1720) are missing.
Author Response
Please sea the attachment

Reviewer 2 Report
This paper reports an interesting clinical observation that hospitalized cancer patients had similar risk of ICU and no excess of risk of death, compared with patients without cancer in the Reggio Emilia Cohort Study. It was also observed that male gender, age > 70 years, metastatic disease bladder cancer and cardiovascular disease were associated with mortality risk in these cancer patients. The results are interesting and important. I have only some minor comments:
- Lines 129-130: What is the number “89.9” in the sentence “The number of cancer patients hospitalized with SARS-CoV-2 infection was 138 (11.3%) for a standardized hospitalization ratio compared to the general population of 89.9 (95%CI 75.5-106.1)”? This needs to clarify.
- In Discussion (lines 235-257), a comparison between the cases in Italy and China is given, while there is no comparison between the cases in Italy and the USA. For the latter, the mortality looks similar between the two countries. This comparison should be included and discussed.
- Lines 292-293: the sentence “Among the 157 patients with solid tumors in Montefiore Health System, 11 (7.0%) had a lung cancer and 11 (55%?) of these died” needs to clarify.
- In Table 5, “ydroxychloroquine” should be corrected to hydroxychloroquine.
Author Response
Please sea the attachment

Reviewer 3 Report
It is a well written manuscript about COVID-19 outbreak in specific Italian region. Although interesting, the data presented endorse or enlarge previous reported data (sometimes short cohorts from different areas of the world do not improve the colelective knowledge)
I have somme minor comments:
1) Please compare your data also with TERAVOLT recently published in Lancet Oncol for putting in context your results with a large and international cohort
2) Discussion is large and lacks of putting into the context your data it seems more a review, please cut the discussion and try to justify your discordant results, obviously endorsing your evidence with bibliography
3) Conclusions should be more concise
Round 2
Reviewer 1 Report
- When continuous variables follow a normal distribution they should be described with the mean and standard deviation.
When continuous variables do not follow a normal distribution they should be described with the median and range.
Therefore, it is not correct to describe a continuous variable by writing the median, range and mean at the same time.
- In addition, authors should indicate the statistical test they have used to consider whether or not the quantitative variables follow a normal distribution.
- The limitations of the study are described in the Conclusions section. It would be better if these limitations were discussed in the Discussion section.
